# The Influence of Broilers’ Body Weight on the Efficiency of Electrical Stunning and Meat Quality under Field Conditions

**DOI:** 10.3390/ani11051362

**Published:** 2021-05-11

**Authors:** Giorgio Smaldone, Stefano Capezzuto, Rosa Luisa Ambrosio, Maria Francesca Peruzy, Raffaele Marrone, Giacomo Peres, Aniello Anastasio

**Affiliations:** 1Department of Agricultural Sciences, University of Naples Federico II, Via Università 100, 80055 Portici, Italy; giorgio.smaldone@unina.it; 2Independent Researcher, 80125 Naples, Italy; stefano.capezzuto@hotmail.it; 3Department of Veterinary Medicine and Animal Production, Unit of Food Hygiene, University of Naples Federico II, Via F. Delpino 1, 80137 Napoli, Italy; mariafrancesca.peruzy@unina.it (M.F.P.); raffaele.marrone@gmail.com (R.M.); anastasi@unina.it (A.A.); 4Official Veterinarian ASL NA 2 nord, 80027 Frattamaggiore, Italy; giacomo.peres@outlook.it

**Keywords:** animal welfare, poultry, stunning process, water bath

## Abstract

**Simple Summary:**

Water-bath stunning represents the most-applied stunning system in poultry slaughtering. This study aimed to assess the efficiency of two types of electrical equipment applied to broilers with different live body weights. Moreover, the influence of the tested stunners on broiler meat quality was evaluated. 6600 broilers, divided into three weight groups, were stunned and the state of unconsciousness and post-mortem defects were evaluated by blinded trained operators. Considering the total body weight, the application of the two stunning systems resulted in a different occurrence of ineffective stunning signs registering statistical differences (*p* < 0.01) among groups. Considering injuries, an inverse relationship between body weight and lesions was found. The results highlighted the effectiveness of both stunning systems that apply the best combination of electrical parameters, taking into account the weight of the animal and ensuring its welfare.

**Abstract:**

Water-bath stunning represents the most-applied stunning system in poultry slaughtering, but within the European Union, specific indications on electric parameters that should be used, such as voltage, are missing. The objective of this study was to evaluate the efficiency of two commercially available types of electrical equipment (A and B) on broilers with different live body weights and the influence of the tested parameters on meat quality. Experimental trials in a European Union-approved slaughterhouse were carried out using two different stunners. 6600 broilers, divided into three weight groups, were stunned applying different protocols based on the same current frequencies and intensity but different voltages. The state of unconsciousness (presence of corneal reflex and wings flapping) and post-mortem defects (pectoral hemorrhages and dark meat) were evaluated by blinded trained operators. The presence of corneal reflex and petechiae were the most reported consciousness signs and post-mortem injuries, respectively. Different weights played an important role within stunner A, registering statistical differences (*p* < 0.01) among groups. Considering injuries, an inverse relationship between body weight and lesions was found. The results highlighted the effectiveness of both stunning systems applying the best combination of electrical parameters considering the weight of the animal and ensuring its well-being.

## 1. Introduction

Animal welfare during stunning and slaughter is a matter of public concern that affects product quality, food safety, and the healthiness of consumers. In this context, there is an increasing interest of stakeholders to develop accurate and efficient techniques able to protect animals during these operations [1,2].

Due to the progressive increase in demand for white meat, nowadays broilers represent one of the most slaughtered animals [3], and therefore the welfare of these animals is an important issue. In European Union (EU), around 80% of broilers are usually stunned through an electrical water bath and only 20% by Controlled Atmosphere Stunning (CAS), whereby individual animals are exposed to gas mixtures [4,5].

During a water bath stunning, several animals are simultaneously immersed in an electrified water bath with a variable capacity. This stunning method is based on the presence of two electrodes, located respectively on the bottom of the tank and on the guideway that transports the animals. The electric circuit is closed when the animal’s head is completely immersed in the stunning tank causing the passage of electric current along the whole body of the animal and thus inducing a state of unconsciousness and insensitivity [6]. These latter states are evaluated through the observation of several scientifically recognized parameters such as changes in animal behavior, physical reflexes, and physiological signs [7,8]. The presence of corneal reflex in broilers is considered a reliable indicator of consciousness in poultry [4,7,9,10,11] and, indeed, if present after stunning may be a sign of brain functions restoring in the animal [12]. Another important sign of consciousness, and therefore incorrect stunning, is the presence of wing flapping; on the contrary, the presence of the wing close to the body is considered a good visual assessment of stunning efficiency [7].

Electrical parameters which should be considered to induce unconsciousness and insensibility are (i) minimum current, (ii) minimum voltage, (iii) frequency of current, (iv) current type (alternating current/direct current), and (v) waveform of the electricity [13]. The wrong choice of electrical parameters or equipment, poor or lack of calibration, the use of low voltage/current or/and the application of high frequencies may represent an animal welfare hazard and may impact meat quality [14]. Electrical stunning values are settled by the EU Regulation N. 1099/2009 [15] which establishes rules on the protection of animals at the time of the killing. However, in Italy until 8 December 2019 with the Italian Ministerial Note of 7 January 2013 [16], the Competent Authority could authorize the use of electrical values lower than those indicated by the EC Regulation No 1099/2009 if animal welfare was respected.

Nevertheless, the respect of welfare is not always associated with meat quality [14]. It is well known that electrical stunning may cause many forms of meat downgrading. In particular, it is acknowledged that using high voltages in a water-bath stunner can lead to poor bleeding [17] but can increase the time of unconsciousness [9]. Among the electrical parameters, frequency is another important factor influencing animal welfare and meat quality. According to the Eu Regulation n. 1099/2009, Food Business Operators (FBO) may use a wide range of frequency values (up to 1500 Hz). High frequencies improve meat quality; however, they have a shorter stun effect on animals, leading to a quick return to consciousness [2,11,18], while low frequencies may cause death but, due to strong muscle contractions, post-mortem defects and hemorrhages may appear [9,19]. For these reasons, an electrical frequency > 300 Hz is usually applied to safeguard the quality of the poultry carcass [20]. However, most of the electrical parameter data for water-bath stunning are laboratory-scale studies and therefore it is difficult to extrapolate directly to large-scale conditions. Hence, this work aimed to assess the efficiency of two types of electrical equipment applied to broilers with different live body weights under field conditions. Moreover, the influence of the tested stunners on broiler meat quality was evaluated.

## 2. Materials and Methods

### 2.1. Animals and Experimental Design

The study was conducted in an approved poultry slaughterhouse in the “Campania region” in southern Italy. A total of 6600 broilers (genetic line Ross 708 six to eight weeks old) were divided into three homogeneous batches (2200 animals per batch) based on their live body weight (group 1 (G1)—live body weight up to 3.4 kg; group 2 (G2)—live body weight up to 3.7 kg; group 3 (G3)—live body weight up to 4 kg). Experimental trials were always performed during the first two hours of work in order to have the same quality of water in the water bath. Each group was slaughtered in a deferred way and broilers were individually hung upside down and immersed into the electrified water bath up to the basal end of their wings for 15 s. The shackle line speed was set at 4000 birds per hour in order to ensure, for each bird, a minimum stunning time of 6 s.

### 2.2. Electrical Stunning Equipment and Protocol

A water bath capable of containing up to 13 broilers and two different electrical stunners was used: each group was split into 2 equal sub-groups and stunned with: (i) stunner A (SA) (Table 1)—Water Stunner BA4 LINCO Food Systems A/S, Aarhus, Denmark- used by way of derogation until the 8 December 2019 due to the lack of a registration system of the stunning process, and; (ii) stunner B (SB) (Table 2)—Cattaruzzi Inhibit Wave 1500, Cattaruzzi S.r.l. san Zeno Naviglio, Brescia, Italy, equipped with an automatic system able to register and change the electrical parameters based on the number of animals in the bath and the their weight.

Sinusoidal alternating current (SAC), frequencies (Hz), and intensity (A) were set in agreement with Annex I, Chapter II, point 6.3 of the EC Regulation 1099/09 and remained fixed during the stunning procedures. The two stunners both worked with the same type of current (SAC). To reach the same intensity, different voltages (V) were applied for stunner A and stunner B. In particular, stunner A required lower voltage than stunner B.

Moreover, for stunner A the voltage was adjusted manually [21] on the stunner control panel and automatically for stunner B, considering the bird’s batch body weight in order to give the same amount of SAC intensity to each group (Table 1 and Table 2). To estimate SAC flowing through each animal, the total electrical current passing through the water bath was divided by the number of broilers present simultaneously in the water bath [22]. A stand-alone ammeter (HT9021 AC/DC TRMS Clamp meter 1000A, HT Italia Srl, Faenza, Italy) was used to confirm the accuracy of the current measurement provided by the stunners and consequently obtain standardization of the stunning process for different poultry categories. No salt was added to the water bath.

### 2.3. Correct Stunning Evaluation

The state of unconsciousness, and therefore the stunning ineffectiveness (IS), was evaluated 20 s after stunning and before bleeding using the following parameters: the presence of corneal reflex (CR) tested by touching the cornea of the bird with a feather to assess blinking, and wing flapping (WF)—in terms of the number of animals without wings close to the body—or both (CRW). The bleed machine was close to the stunner exit in order to avoid the return of consciousness of broilers.

### 2.4. Post-Mortem Evaluation

After slaughtering, all birds were scalded, mechanically plucked, eviscerated, and held overnight at 5 °C and, subsequently, examined to determine the frequency of post-mortem defects (PMD): (i) bruising—pectoral hemorrhages (PH), intended as branched appearance due to extravasating blood that followed the direction of the muscle fiber as reported by Kranen [23], and (ii) dark meat (DM) as purplish-cyanotic coloration imparted to the skin, mucous membranes and muscle by poorly oxygenated hemoglobin on *Pectoralis major*. Both stunning and carcass evaluations were done by trained operators in blind.

### 2.5. Physical-Chemical Analysis

Both pH and Water Holding Capacity (WHC) were evaluated in the *Pectoralis major* in all carcasses PMD and in 100 randomly selected carcasses without any sign of post-mortem defects (NPMD). Measurement of pH was performed 24 h postmortem using a portable pH-meter (Crison PH25, Barcelona, Spain) by inserting electrodes into the muscle. WHC was evaluated as described by Carvalho [24]: briefly, 24 h postmortem about 10 g of meat, collected from each animal, were cut into cubes 1.0 ± 0.01 g and carefully placed on acrylic plates between 2 pieces of filter paper (No. 4; Whatman International Ltd., Maidstone, UK) and then left under a weight of 1 kg for 5 min; then they were weighed and WHC was determined using the following equation:100 − [(W_i_ − W_f_/W_i_) × 100]
where W_i_ and W_f_ are the initial and final sample weight, respectively. All samples were analyzed in duplicate.

### 2.6. Statistical Analysis

Differences in the occurrence (%) of IS (CR, WF, and CRW) signs and PMD (PH and DM) by using the two stunners (SA and SB) for each weight group were assessed by the two-sided chi-square test (MedCalc for Windows, version 18.11.3—MedCalc Software, Ostend, Belgium). The effect of the different voltages applied in different weight groups on IS and PMD was evaluated with a cross-tabulation by chi-square test (MedCalc for Windows, version 18.11.3—MedCalc Software, Ostend, Belgium). Correlation between IS and PMD and between pH and WHC were determined by Pearson linear correlation coefficient (both per stunners and group). Differences of pH and WHC values by using the two stunners (SA and SB) and the differences of WHC values between carcasses showing PMD and carcasses without any sign of post-mortem defects were assessed using student’s *t*-test (MedCalc for Windows, version 18.11.3—MedCalc Software, Ostend, Belgium). For all tests, a probability value of <0.05 (*p* < 0.05) was defined as statistically significant.

### 2.7. Ethical Approval

The request for ethical approval has been sent to Ethical Animal Care and Use Committee University of Naples Federico II: the experimental trials result excluded from the Directive 2010/63/EU on the protection of animals used for scientific purposes (Università Degli Studi di Napoli Federico ii, Centro Servizi Veterinari PG/2020/0089492 del 30/10/2020).

## 3. Results

### 3.1. Correct Stunning Evaluation

A total of 67 animals (1.01%) showed signs of stunning ineffectiveness (Table 3).

Among the IS parameters, the presence of corneal reflex was the most-frequently detected (*n =* 34, 50.7%). By device used, broilers stunned with SA and SB exhibited incorrect stun signs of 1.06% (*n =* 35) and 0.96% (*n =* 32), respectively, although no significant differences between the two devices were observed. WF was present only in 16 animals (0.244%) in both groups. Considering the total body weight, the application of the two stunning systems resulted in a different occurrence of the IS signs. Different weights played an important role within system A, registering statistical differences (*p* < 0.01) among groups. In particular, with SA the application of low voltages (53–80 V) caused stunning ineffectiveness rates inversely related to body weight, ranging from 1.18 to 0.90% (G1 and G3, respectively). On the contrary, by using SB, the occurrence of IS signs was not related to the weight and the highest percentage of IS signs was observed in the broiler with the highest body weight (G3). Regardless of the stun system, G1 groups accounted for the highest percentages of the presence of corneal reflex, and animals with live body weight up to 4 kg (G3) registered the highest frequency (*p* < 0.05) of the simultaneous presence of the corneal reflex and wings flapping (CRW).

### 3.2. Post-Mortem Evaluation

A total of 94 animals showed post-mortem defects (Table 3). Among PMD, pectoral hemorrhages were the lesions most frequently observed (*n =* 74, 79.86%—Figure 1 and Figure 2).

Based on the device used, SA caused the highest number of PMD (*n =* 54, 1.63%) and, considering the different weight groups, both SA and SB were less effective on G1 (*n =* 26 and *n =* 15 respectively). Similar results were observed examining each stunning system individually, reinforcing the inverse relationship between body weight and evident lesions. Although a positive correlation was present (R = 0.2298), the relationship between IS and PMD was not statistically significant (*p* > 0.05) and broilers with wings flapping did not present PMD.

### 3.3. Physical-Chemical Analysis

In Table 4, pH and WHC (%) mean values (± SD) are reported.

The lowest mean WHC values were recorded for PMD breast samples: significative statistical differences in WHC were observed between the SA and SB (*p* < 0.05) and between breast with lesions and without lesions (*p* < 0.05). Differently, no statistical differences in pH values were recorded between SA and SB and between breast fillets. Finally, no significant correlation between pH and WHC was found (*p* > 0.05).

## 4. Discussion

Taking into account the stunning time, the main electrical parameters that can be modified in order to improve the stunning effectiveness are frequency, intensity, and voltage. According to the Eu Regulation n. 1099/2009, FBO may use a wide range of frequency values (up to 1500 Hz) and a current intensity, ranging from 100 to 200 mA. FBO may use, therefore, different combinations of electrical parameters in order to safeguard both animal welfare and the possible downgrading of meat minimizing economic losses [2,8,25,26]. The amount of the current has great importance on stunning efficiency, and it is proven that currents below 100 mA do not reach an adequate level of unconsciousness [10]. The resistance to the passage of electric current is variable according to several factors such as the number of animals in the water bath, their gender, body size and weight, muscle growth, fat content [10,11,27,28], and plumage conditions (wet, dry or dirty) [29]. Even though most of the authors based their research on the modification of frequency, in the present work this parameter was fixed and set at 1400 Hz, while voltage was adjusted following batch weights to achieve a SAC intensity of about 200 mA/ subject, in accordance with Reg (EC) n. 1099/2009 CE safeguarding animal welfare [30]. However, depending on the applied value there could be some inconveniences and/or advantages.

In the present study, a low percentage of animals (around 1%) showed signs of stunning ineffectiveness. It has been previously reported that when low intensities are applied, high electrical frequencies are less effective in overcoming impedance in the tissues of the birds leading to a short period of unconsciousness in the animals [2,22]. The outcomes of the present work confirm the efficacy of high frequency combined with the high root mean square current up to 200 mA as already reported by Raj [25] and Girasole [2]. This latter aspect is confirmed also by the studies of Karunanayaka [31] and Ciobanu [32] which demonstrated that the use of high frequencies speeds up the stunning process in broilers and, concerning the intensity of the current, makes stunning more effective. Based on these considerations, the combinations of stunning electrical parameters proposed appear effective to induce suppression of stunning ineffectiveness.

Among signs of stunning ineffectiveness, the presence of corneal reflex was the most frequently detected in this study. By using lower frequencies, other authors reported higher percentages of animals with a positive corneal reflex [22,33]. The eye reflex was recognized by several authors as the physical reflex most reliable in evaluating unconsciousness [10,11], although its absence does not necessarily indicate the absence of pain. In agreement with Prinz [10], in the present study, the evaluation of WF was also assessed as an indicator for stunning effectiveness because it is easy to observe, and it makes results highly reliable [34]. Contrary to corneal reflex, WF was present in few animals (0.244%). This result was in contrast with Grilli [35] that, in a trial conducted on more than 23,000 animals, found an average WF value of 4.9%. The authors proposed the simultaneous evaluation of both corneal reflex and wings flapping to better describe the consciousness state. Regarding electrical parameters as intensity and frequency, Girasole et al. [2] registered a 12% of CR in poultry stunned with 200 mA and low frequency of 750 Hz, without compromising of carcass and meat. However, Hindle [36] noted that the increase in intensity must correspond to a rise in frequency in order to correctly evaluate the suppression of CR, describing a scenario similar to those tested in this study. Nevertheless, Prinz [11] demonstrated that increasing the frequency to 1500 Hz does not ensure a low percentage of CR in poultries stunned, results that appear in contrast with the present work. Several authors [35,37] suggest post-mortem inspection to detect incorrect catching, handling, and stunning procedures. For instance, epileptic seizures and ventricular fibrillation, as a consequence of ineffective stunning, may cause breast injuries and of downgrading of meat. According to this concept, Goksoy et al. [37] stated that about half of broilers had ventricular fibrillation when stunned by applying a high voltage and low-frequency electric current. Similar consideration was expressed by Raj [38] who reported that about 90% of chickens subjected to electric stunning with low frequencies and high intensity had epileptic seizures. In our study, low rates of postmortem defects were detected (1.42%), demonstrating that high and constant electrical frequency coupled with a manual adjustment of voltage can positively impact not only on the welfare of poultry (Table 3) in the stunning phase but also on the final quality of the meat (*p* < 0.05). Results were in agreement with Xu [39] who asserted that the use of high electrical frequency had a positive effect on the texture of the meat, with Gregory [40] who affirmed that the application of high-frequency electric currents (1500 Hz) induced fewer hemorrhages on meat compared to low frequency and with Girasole [2] that state that post-mortem lesions decrease as the stunning frequency and intensity increase, with a reduction of up to 100% for very high-frequency levels (>1200 Hz).

Electrical parameters could also influence pH and WHC, which affect the important added value of meat and meat products such as appearance, production efficacy/profitability, and consumption quality. Lower mean WHC values were recorded for post-mortem defects breast samples confirming the findings of Northcutt [41]. Low WHC in chicken meat is a predisposing factor of PSE meat. Takahashi [42] reported higher WHC in breast meat of broilers stunned with high frequencies. Our results, in line with findings of Siqueira [33] and Karunanayaka [31], confirm that PMD breast samples have lower WHC than normal ones.

The body weight plays an important role in the stunning efficiency because of the large variation in resistance to the current flow among animals with different percentages of fat content. In the present work, the occurrence of signs of stunning ineffectiveness and post-mortem defects were different among birds having different live body weights. In particular, the live body weight plays an evident role in the onset of post mortem defects in both stunning systems tested: G1 groups have more defects than G2 and G3. Moreover, pectoral hemorrhages and dark meat investigated showed a perfect trend overlapping which confirms the weight of different percentages of fat content on the variation in resistance to the current flow. However, the results of this study are in contrast with Rawles [27] that found no differences between weight groups. It is worth noting that the results of the present study displayed a not-homogeneous response connected to body weight, being influenced also by voltage adopted. Indeed, only when voltage values higher than 150 were adopted, did animals with a live body weight up to 4 kg show the highest percentage of signs of stunning ineffectiveness. The low voltage ranged from 53 to 80, and did not significantly affect the effectiveness of stunning systems on the base of live body weight.

## 5. Conclusions

The present work aimed to assess the efficiency of two types of electrical equipment applied to broilers with different live body weights under field conditions. Moreover, the influence of the tested stunners on broiler meat quality was evaluated. Based on results, the application of high frequencies, coupled with high intensity and manual voltage adjustment, guarantee a high level of unconsciousness of the birds and a low incidence of injuries of the final product. Therefore, high frequencies combined with high voltage should be applied by FBO during the stunning process. However, according to data, the occurrence of signs of stunning ineffectiveness and post-mortem defects could also be affected by the weight of the animals. However, alternative slaughter methods in combination with a low electrical stunning should be investigated to hopefully eliminate the conflict between animal welfare, meat quality, and safety of workers’ health.

## Figures and Tables

**Figure 1 animals-11-01362-f001:**
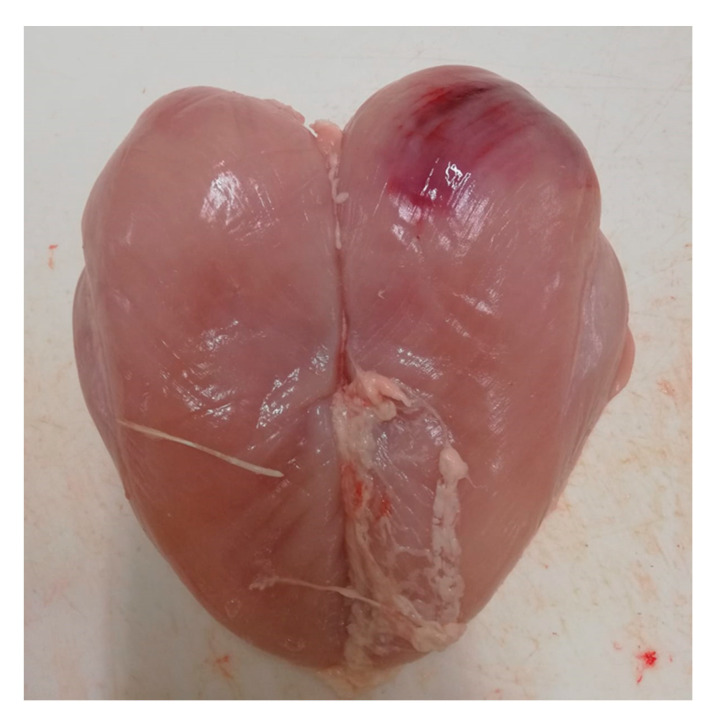
Presence of post-mortem defects on *Pectoralis major*: pectoral hemorrhages.

**Figure 2 animals-11-01362-f002:**
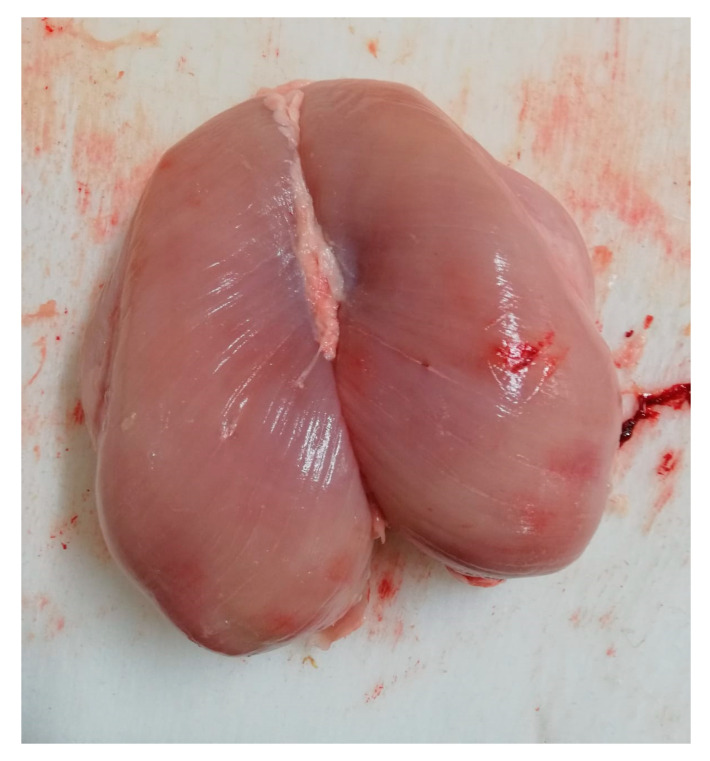
Presence of post-mortem defects on *Pectoralis major*: dark meat and pectoral hemorrhages.

**Table 1 animals-11-01362-t001:** Work parameters of stunner A.

Weight Group	Frequencies [Hz]	Intensity [mA ± SD]	Voltage [V]	Intensity/Broiler [mA ± SD]
G1	1400	2712.4 ± 7.31	53	208.65 ± 0.56
G2	1400	2720.7 ± 12.19	76	209.28 ± 0.93
G3	1400	2729.0 ± 17.07	80	209.92 ± 1.31

**Table 2 animals-11-01362-t002:** Work parameters of stunner B.

Weight Group	Frequencies [Hz]	Intensity [mA ± SD]	Voltage [V]	Intensity/Broiler [mA ± SD]
G1	1400	2720.7 ± 12.19	150	209.28 ± 0.93
G2	1400	2775.5 ± 14.63	170	213.5 ± 1.12
G3	1400	2791.5 ± 24.39	190	214.73 ± 1.87

**Table 3 animals-11-01362-t003:** Percentage of stunning ineffectiveness (IS) (also expressed in percentage of presence of corneal reflex (CR), the wings flapping (WF) or both (CRW)) and Post Mortem Defects (PMD) (pectoral hemorrhages (PH) and dark meat (DM)). Statistical analysis was performed comparing each group for each parameter. Different superscript uppercase letters indicate a significant difference at *p* < 0.01. Different superscript lowercase letters indicate a significant difference at *p* < 0.05. a and b were used for comparison between the two different stunning systems within the same weight group. x and y were used for comparison among weight groups stunned with the same protocol.

Stunner	Weight Group	Total Number of Animals (n)	IS (%)	CR (%)	WF (%)	CRW (%)	PMD (%)	PH (%)	DM (%)
SA	G1	1100	1.18	0.90 ^X^	0.18	0.09 ^x^	2.36 ^x^	1.72 ^x^	0.63
SA	G2	1100	1.09	0.72 ^X^	0.36	0 ^X^	1.45	1.09	0.36
SA	G3	1100	0.90	0 ^Y^	0.18	0.72 ^yY^	1.09 ^y^	0.72 ^y^	0.36
SB	G1	1100	1.00	0.54	0.45 ^x^	0 ^X^	1.36	1.09	0.27
SB	G2	1100	0.45 ^x^	0.45	0 ^y^	0 ^X^	1.18	1.09	0.09
SB	G3	1100	1.45 ^y^	0.45	0.27	0.72 ^Y^	1.09	1.00	0.09
SA	Total	3300	1.06	0.54	0.24	0.27	1.63	1.18	0.45 ^a^
SB	Total	3300	0.96	0.48	0.24	0.24	1.21	1.06	0.15 ^b^

**Table 4 animals-11-01362-t004:** WHC and pH of poultry meat measured on carcasses with (PMD) and without post mortem defects (NPMD) after stunning with stunner A (SA) and stunner B (SB). Statistical analysis was performed as follows: ^A, B^ values within a stunner and between PMD/NPMD differ significantly; ^X, Y^ values between PMD/NPMD belong to different stunners. Different superscript uppercase letters indicate a significant difference at *p* < 0.01. Different superscript lowercase letters indicate a significant difference at *p* < 0.05.

Parameters	SA PMD	SA NPMD	SB PMD	SB NPMD
pH	5.88 ± 0.15	5.90 ± 0.11	5.87 ± 0.13	5.86 ± 0.13
WHC	79.58 ± 1.19 ^XA^	81.31 ± 1.70 ^Ya^	80.91 ± 1.48 ^XB^	82.15 ± 1.07 ^Yb^

## Data Availability

The data presented in this study are available on request from the corresponding author.

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
