# Peer review of "The Influence of Broilers’ Body Weight on the Efficiency of Electrical Stunning and Meat Quality under Field Conditions"

_animals, 2021, doi:10.3390/ani11051362_

Round 1

Reviewer 1 Report

in the Materials and methods, it is explained that the settings of the stunning equipment are such, that a standardised SAC is conducted through the broilers. However, it is not very clear how the value f around 200 mA was arrived at. How was it decided that this was the current needed?

High frequency AC does prevent damage, but there are also papers that indicate that high frequencies are less effective in overcoming impedance in the tissues of the birds, thus requiring higher voltages to achieve sufficient current. This may need to be addressed as well.

The subject of study, in its current form, is to assess the effect of 200 mA of current on the stunning effect and the meat quality in broilers of different weight categories. This is a legitimate subject, but it would have been beneficial if at least in the discussion some attention was given to the fact that in an average abattoir the weight of the birds does not trigger the amount of current going through each bird, as birds with a variety of weights go through the same water bath. In practice, the currents going through each bird will therefore vary greatly. The point is made that adding an ammeter would help in the assessment of the current going through each bird. While this is true, it does not in itself improve welfare, as it will only indicate whether that bird has received enough current in that instance. Some discussion needs to be given to how this improves welfare.

Overall, the implications of the results of the study for the practical application of stunning could get a bit more attention in the discussion. What is the take home message that all FBO's should take to heart, irrespective of the stunning equipment they are using, and the frequency they are using? 

Author Response

Reviewer 1

Reviewer: in the Materials and methods, it is explained that the settings of the stunning equipment are such, that a standardised SAC is conducted through the broilers. However, it is not very clear how the value f around 200 mA was arrived at. How was it decided that this was the current needed?

Authors: Thank you to point this out. To estimate SAC flowing through each animal, the total electrical current passing through the water bath was divided by the number of broilers present simultaneously in the water bath. Please refer to lines:128-130. The authors decided to set up the current value in agreement with EU Regulation N. 1099/2009 which establishes rules on the protection of animals at the time of the killing. For frequency up to 1500 Hz, for each broiler, a minimum of 200mA must be guarantee. This value was confirmed by the stand alone ammeter.

Reviewer: High frequency AC does prevent damage, but there are also papers that indicate that high frequencies are less effective in overcoming impedance in the tissues of the birds, thus requiring higher voltages to achieve sufficient current. This may need to be addressed as well.

Authors: Based on the reviewer’s suggestion the sentences: It has been previously reported that when low intensities are applied, high electrical frequencies are less effective in overcoming impedance in the tissues of the birds leading to a short period of unconsciousness in the animals.” Please refer to lines 252-254

Reviewer: The subject of study, in its current form, is to assess the effect of 200 mA of current on the stunning effect and the meat quality in broilers of different weight categories. This is a legitimate subject, but it would have been beneficial if at least in the discussion some attention was given to the fact that in an average abattoir the weight of the birds does not trigger the amount of current going through each bird, as birds with a variety of weights go through the same water bath. In practice, the currents going through each bird will therefore vary greatly. The point is made that adding an ammeter would help in the assessment of the current going through each bird. While this is true, it does not in itself improve welfare, as it will only indicate whether that bird has received enough current in that instance. Some discussion needs to be given to how this improves welfare.

Authors: As stated above, frequency and intensity have been set in agreement with EU Regulation N. 1099/2009 which establishes rules on the protection of animals at the time of the killing. For frequency up to 1500 Hz, for each broiler, a minimum of 200mA must be guarantee. In most poultry slaughterhouses broilers arrive in large batches of the same weight and, depending on this, the stunning parameters are set. Modern stunners, as stunner B of this study, records the weight of each hung-up broiler and give the current based on the mean weight. It’s unusual that in the same moment are present in water-bath broilers with different weight and, for this reason, there’s no possibility of incorrect stunning and therefore, lack of welfare.

Reviewer: Overall, the implications of the results of the study for the practical application of stunning could get a bit more attention in the discussion. What is the take home message that all FBO's should take to heart, irrespective of the stunning equipment they are using, and the frequency they are using? 

Authors: Thanks to the reviewer's comments the conclusion has now been changed as: “The present work aimed to assess the efficiency of two electrical equipment applied to broiler with different live body weights under field conditions. Moreover, the influence of the tested stunners on broiler meat quality was evaluated. Based on results the application of high frequencies, coupled with high intensity and manual voltage adjustment guarantee a high level of unconsciousness of the birds and a low incidence of injuries of the final product. Therefore, high frequencies combined with high voltage should be applied by FBO during the stunning process. However, according to data, the occurrence of signs of stunning ineffectiveness and post-mortem defects could be also affected by the weight of the animals. However, alternative slaughter methods in combination with a low electrical stunning should be investigated to hopefully eliminate the conflict between animal welfare, meat quality, and safe from a worker’s health.”Please refer to lines 321-329

Reviewer 2 Report

Content. This paper has low significance because the authors have not paid attention to the fundamentals of electric stunning.  The key aspect of the paper seems to be a comparison between two stunning situations SA and SB. Stun currents are similar but the voltages are remarkably different with the apparent impedances of the birds in the two stunners differing by a factor of almost 3. This huge difference should have immediately sparked questions. What is causing this difference since it is an implicit assumption of this paper that the birds are the same.  Without understanding the reason for the difference no further progress can be made and all conclusions are meaningless. Such variation would not normally be found in a water-bath stunner.

It could be of interest that the authors find higher levels of unconsciousness than other authors (line 250). However, for this difference to have significance we need to be sure that the assessment method was comparable. The tests are relatively subjective and the details of how and when the test is done and assessed could have a very significant impact on the finding. I see no such examination and so the reader is left wondering if this is a difference in methodology or result.

The conclusion opens with the statement that “The present work aimed to find an optimal set of electrical parameters for the water bath stunning ……..” This is a very different aim to that presented in the introduction, and one not remotely achieved.  

Comments on presentation.  This paper is difficult to read because

  • It uses far too many acronyms. Please try to reword the text so it used normal English language rather than this abbreviation shorthand. You need remember that as an author you have lived with this work for weeks so every abbreviation is logical and familiar. In contrast, the reader spends ten minutes or an hour reading it and so is faced with a huge number on new abbreviations that need constantly checking.
  • The text lacks structure. The discussion section is nearly 90 lines of text unbroken by paragraphs. Please try to break your text up into discrete thoughts which you introduce, justify and conclude. Then start a new paragraph for a new thought. This process will help you order your thoughts and will help the reader to discern what you wish to communicate.
  • The discussion is not focused. Large amounts of the discussion seem to be unrelated to the results presented in this paper. Please try to keep the discussion focused on the results of the experiments, their significance and their implications.
  • The text would be improved with the assistance from a native English speaker, however points 1 and 2 are a far more significant barrier to comprehension.

Other. I have not noted typgraphical errors because the points made above are of far more importance, however, I noticed in passing that the equation in line 154 is missing parentheses and the link to reference 4 reports an error.

Author Response

Reviewer 2

Reviewer: This paper has low significance because the authors have not paid attention to the fundamentals of electric stunning.  The key aspect of the paper seems to be a comparison between two stunning situations SA and SB. Stun currents are similar but the voltages are remarkably different with the apparent impedances of the birds in the two stunners differing by a factor of almost 3. This huge difference should have immediately sparked questions. What is causing this difference since it is an implicit assumption of this paper that the birds are the same.  Without understanding the reason for the difference no further progress can be made and all conclusions are meaningless. Such variation would not normally be found in a water-bath stunner.

Authors: The authors thank the reviewer to point this out: under the multiple bird water bath stunning system, all the birds passing through the electrified water bath will be exposed to a constant voltage, and the electrical flow through each bird will be dependent on impedance caused by each bird. In this study, in order to stun n. 3 homogeneous bird batches, two different stunners were used: in the case of Stunner A, voltage was adjusted manually to try to achieve a stunning current of about  200 mA/bird (about 2700 mA measured and recorded on stunner control panel). In case of Stunner B, voltage was adjusted automatically by equipment considering the mean bird's batch body weight (line 127). The application of different voltages in two stunners for the same bird batch is linked to the different equipment: stunned A needs lower voltage than stunner B to achieve the same intensity of current. The additional stand alone ammeter confirmed the amount of current passing in the water bath. This has now been included into the manuscript. Please refer to line 123-127

Reviewer: It could be of interest that the authors find higher levels of unconsciousness than other authors (line 250). However, for this difference to have significance we need to be sure that the assessment method was comparable. The tests are relatively subjective and the details of how and when the test is done and assessed could have a very significant impact on the finding. I see no such examination and so the reader is left wondering if this is a difference in methodology or result.

Authors: The authors agree with the reviewer. Therefore, when differences or similitudes were observed, the method of each research was considered.

Reviewer: The conclusion opens with the statement that “The present work aimed to find an optimal set of electrical parameters for the water bath stunning ……..” This is a very different aim to that presented in the introduction, and one not remotely achieved.

Authors: Based on reviewer suggestion the conclusions have been re-written as: “The present work aimed to assess the efficiency of two electrical equipment applied to broiler with different live body weights under field conditions. Moreover, the influence of the tested stunners on broiler meat quality was evaluated. Based on results the application of high frequencies, coupled with high intensity and manual voltage adjustment guarantee a high level of unconsciousness of the birds and a low incidence of injuries of the final product. Therefore, high frequencies combined with high voltage should be applied by FBO during the stunning process. According to data, the occurrence of signs of stunning ineffectiveness and post-mortem defects is largely affected by the weight of the animals. However, alternative slaughter methods in combination with a low electrical stunning should be investigated to hopefully eliminate the conflict between animal welfare, meat quality, and safe from a worker’s health..” Please refer to lines 321-329

Comments on presentation.  

Reviewer: This paper is difficult to read because:

Reviewer: It uses far too many acronyms. Please try to reword the text so it used normal English language rather than this abbreviation shorthand. You need remember that as an author you have lived with this work for weeks so every abbreviation is logical and familiar. In contrast, the reader spends ten minutes or an hour reading it and so is faced with a huge number on new abbreviations that need constantly checking.

Authors: Thank you for the suggestion; the necessary corrections were made in the manuscript

Reviewer: The text lacks structure. The discussion section is nearly 90 lines of text unbroken by paragraphs. Please try to break your text up into discrete thoughts which you introduce, justify and conclude. Then start a new paragraph for a new thought. This process will help you order your thoughts and will help the reader to discern what you wish to communicate.

Authors: Thank you for the suggestion. The discussions were re-written based on reviewer comments.

Reviewer: The discussion is not focused. Large amounts of the discussion seem to be unrelated to the results presented in this paper. Please try to keep the discussion focused on the results of the experiments, their significance, and their implications.

Authors: Thank you for the suggestion; the discussions were implemented

Reviewer: The text would be improved with the assistance from a native English speaker, however points 1 and 2 are a far more significant barrier to comprehension.

Authors: Thank you for the suggestion; a language check was made

Reviewer: Other. I have not noted typgraphical errors because the points made above are of far more importance, however, I noticed in passing that the equation in line 154 is missing parentheses and the link to reference 4 reports an error.

Authors:  Thank you for the suggestion; the link of reference 4 and the equation in line 160 has been fixed.

Round 2

Reviewer 2 Report

The authors have not materially improved this paper. The authors have still not enquired about the reason for the remarkable difference in the apparent impedance of the birds. Where the birds in one group appear to have an impedance more than twice that of the other group there must be a significant factors that can be isolated, understood and reported. Is it the wave form? Is it the depth to which the birds are inserted into the waterbath, is it the electrode arrangement? Is it the shackle design? My suspicion is that the difference is due to error in the current measurement technique. Do the authors understand true nature of the wave form and the consequences of this for measurement? The almost complete absence information about the voltage signals suggests a lack of understanding or of enquiry. However whether this guess is right or wrong the authors undermine their credibility by failing to see and respond to this elephant sized factor around which the whole paper pivots. Rectifying this omission is not complicated. It does not need the whole trial to be re-run. A competent engineer could inspect these machines in a processing factor and identify or eliminate potential reasons within a few hours.

Author Response

Dear reviewer,

Thanks for your comments, answers in PDF file. 

Kind regards
